# Directed Evolution Methods for Enzyme Engineering

**DOI:** 10.3390/molecules26185599

**Published:** 2021-09-15

**Authors:** Saurabh Rajendra Nirantar

**Affiliations:** Illumina Singapore Inc., Singapore 757716, Singapore; snirantar@illumina.com

**Keywords:** directed evolution, enzyme engineering, protein engineering, high throughput screening

## Abstract

Enzymes underpin the processes required for most biotransformations. However, natural enzymes are often not optimal for biotechnological uses and must be engineered for improved activity, specificity and stability. A rich and growing variety of wet-lab methods have been developed by researchers over decades to accomplish this goal. In this review such methods and their specific attributes are examined.

## 1. Introduction

Enzymes catalyze all the fundamental transformations required to convert matter and energy into living systems. Society has benefited profusely from adaptation of pre-existing enzymes for industrial, research and medical uses [1,2,3,4,5,6]. However, pre-existing enzymes are often not optimal for industrial applications due to temperature, pH and solvent stability, as well as substrate specificity and activity limitations [7,8,9,10].

To overcome these limitations, researchers have been developing methods to replicate evolutionary processes to enable improvement of enzymes’ properties as required. These methods constitute a critical toolbox, whose importance for public welfare was recently recognized by the awarding of the Nobel Prize for Medicine in 2018 to Frances Arnold, and are reviewed in this article.

## 2. Generation of Diversity to Explore Sequence Space

The first step for any enzyme evolution campaign is to create a library of mutants of the target enzyme. These mutants sample sequence space in which it is hoped a satisfactory solution will be present, which will be uncovered by the process employed for screening. Libraries can broadly be of two types, targeted or random.

Targeted libraries mutagenize only a region of interest or specific amino acid positions of the gene, to increase the odds of finding a desirable mutant. The region of interest may be identified based on a crystal structure or homology model which can be used to infer amino acid positions important for substrate binding or catalysis. Targeted libraries are of particular interest when seeking to improve properties which are disproportionately determined by a few positions, such as substrate specificity. Targeted libraries can be constructed by using site directed mutagenesis using oligos with degenerate codons like nnk or nns at the site of interest [11,12,13]. An attractive alternative to degenerate codons are Trimer codon carrying oligos, which use an equimolar mix of trimeric phosphoramidates codingfor optimal codons (usually for *E. coli*) for 19 or 20 amino acids [14]. This avoids skewed representations or rare codons or stop codons. Such Trimer phosphoramidite mixes can also be ordered customized from certain vendors such as IDT. Another option is to have the entire library constructed from custom vendors like Twist or Genscript.

Random libraries, in contrast, target the whole gene of interest and are more relevant when seeking to improve globally determined properties like thermal stability or where detailed information about the enzyme of interest is not available. Such libraries can be made easily using traditional error prone PCR [15]. One shortcoming of error prone PCR is that as a given codon will most likely have only one mutation, a given amino acid can only be mutagenized to a few others rather than all 19 other possibilities. For instance, a wild type GGT codon is highly unlikely to yield an ATG or TTT codon using random mutagenesis as more than one mutation would be required in the same codon. Under such circumstances, custom DNA library vendors can provide a “scanning” library which contains all 20 amino acids at each position, one mutation at a time, which might be preferable to in-house construction. Such libraries can then be combinatorially shuffled to yield a library with a greater sampling of the mutational space.

Other possibilities and variations for library generation include mutagenic strains for continuous mutagenesis [16,17], shuffling of multiple single mutants to create synergistically improved combined mutations [18].

## 3. Library Screening Methods

Once the library is designed and synthesized, it must then be screened to identify mutants with desired properties. The first step, of course, is to express the library and create a pool of protein mutants for an assay (the phenotype), each of which must retain a traceable connection to the gene mutant (the genotype) from which it was derived. This is because only the gene can be amplified and sequenced and is therefore required to enable recovery and identification of the useful protein mutant.

This review classifies screening methods into two broad categories: those based on optical signal detection and those based on survival, retrieval or growth advantages conferred on the host by the mutant enzyme. The former is generally more quantitative but requires more instrumentation as well as the establishment of an optical assay, while the latter is generally simpler to implement but is more qualitative or semi-quantitative. Optical methods have been further classified into micro scale vs. macro scale methods. Within each category, there are many different approaches, which are described in more detail below.

### 3.1. Optical Methods

#### 3.1.1. Optical Methods > Micro-Scale Methods

All other factors being equal, it is generally preferable to adopt a screening method which uses a small reaction scale, as this enables lower reagent costs as well as the option of screening large numbers to find rare high performing mutants. The use of emulsions to generate isolated picoliter volume droplets, each acting as an independent reactor, has gained popularity since this technology was first described by Tawfik and Griffiths [19]. They used an emulsified in vitro transcription translation (IVTT) mix containing a DNA template coding for HaeIII methylase followed by a HaeIII site and thereafter a biotin moiety. Only a gene coding for a functional HaeIII methylase would methylate its site and prevent subsequent restriction by HaeIII restriction endonuclease, thereby enabling pulldown via the biotin moiety.

##### Optical Methods > Micro Scale Methods > FACS Based Double Emulsion Methods

Subsequently, other researchers have introduced many further refinements, such as the use of fluorogenic substrates, the encapsulation of whole cells expressing enzyme mutants (which allows for enzyme buffers other than those needed for IVTT—a limitation of the original paper mentioned above), and the use of water-in-oil-in-water “double emulsions”, whose aqueous outer phase allows for FACS (Fluorescence Activated Cell Sorting) sorting, thus enabling rapid, quantitative isolation of desirable mutants from libraries of millions and more. Bernath et al. demonstrated that heterogenous bulk double emulsions could be sorted on FACS and lead to enrichment of the correct moiety [20]. Later, Mastrobattista et al. used in vitro translation of a mixture of mutant genes along with a fluorigenic substrate in water-in-oil emulsion, followed by double emulsification, to create a continuous aqueous phase, followed by FACS sorting of these double emulsions to evolve β-galactosidase activity in Ebg, a protein of unknown activity [21]. Around the same time, Aharoni et al. used a library of >10^7^ *E. coli* cells expressing mutants of serum paraoxonase (PON1) encapsulated in a double emulsion and sorted by FACS to discover a variant with 100× improved activity [22]. The double-emulsion followed by FACS approach has become progressively popular and technologically sophisticated (see schematic in Figure 1A), with microfluidic systems capable of generating uniform monodisperse droplets, and numerous applications to different enzymes have been demonstrated [23,24,25,26].

##### Optical Methods > Micro Scale Methods > Water-in-Oil Emulsion Sorting Methods

More recently, researchers have demonstrated the construction of microfluidic devices which allow Fluorescence Activated Droplet Sorting (FADS) using single (water-in-oil) emulsions (see Figure 1B for an example). Such devices allow for direct sorting of water-in-oil emulsions but generally have to be constructed in-house and may not match the sorting rate as well as the variety of fluorescence options available with commercial FACS instruments used with water-in-oil-in-water double emulsions. Ahn and Kerbage first demonstrated a microfluidic device able to sort water in oil emulsions [27]. Baret et al. [28] later demonstrated a device able to sort water in oil droplets at upto 2000 Hz, which could sort droplets containing β-galactosidase containing cells from an excess of droplets without such activity. Agresti and co-workers [29] used a device which could sort water-in-oil droplets containing yeast cells displaying variants of Horseradish Peroxidase (HRP). Based on the intensity of fluorescence of HRP turnover products, they successfully evolved significantly faster variants. Other groups have also demonstrated the use of such devices to evolve a variety of enzymes [30,31,32,33]. Interestingly, Gielen et al. [34] have demonstrated an absorbance based FADS device instead of the more common fluorescence readout. These methods now routinely allow for the quantitative sorting of well over 10^7^ variants.

One limitation of the above-described emulsion methods is the inability to add or wash away reagents, limiting the read-out to cases where one-step fluorescent assays are available. Recent work has demonstrated the addition of reagents at a later stage by controlled merger of droplets. For instance, Holstein et al. [35] recently demonstrated the selection of Savinase protease using a sophisticated microfluidic emulsion device capable of providing new reagents as required by controlled droplet merger. Other examples of devices capable of adding new reagents by droplet merger are reviewed by Weng and Spoonamore [36]; however, widespread use of such devices for enzyme evolution is still some distance away.

##### Optical Methods > Micro Scale Methods > Matrix Capture Followed by FACS

Other workers have developed FACS methods which allow for multiple steps of addition of new reagents and washing before assay. These methods use different matrices, such as Streptavidin beads or agarose, to capture genes and their cognate proteins on or in the same bead, followed by a fluorescent assay whose outcome is also capture locally, allowing for genotype-phenotype linkage followed by sorting. For example, Griffiths and Tawfik [37] developed a multistep method which uses Streptavidin coated beads to successively capture a gene, then its cognate protein (in emulsion via biotinylated antibodies), and finally a substrate of the said protein, whose turnover can be detected using fluorescent antibodies followed by FACS. This method has been applied by other researchers to different enzymes [38,39]. Other researchers have used inducible gelling agents like agarose to serve the same purpose (see schematic in Figure 1C) [40,41].

##### Optical Methods > Micro Scale Methods > Well Arrays Combined with Microscopy

An alternative format to emulsions is the use of microwell or microcapillary arrays. Like emulsions, these arrays physically retain the genotype and phenotype. Instead of a FACS instrument, however, a microscope is used to image the array. Unlike emulsions, these arrays can be repeatedly imaged to extract kinetic information; however, selective recovery is less convenient than with emulsions. Cochran’s group demonstrated a microcapillary array with millions of individual capillaries, which stochastically captures individual yeast cells displaying protein mutants, which turn over a fluorescent substrate. After assay, live cells are recovered by using a laser pulse focused on the correct capillary which expels the contents thereof (Figure 1D). The authors demonstrated the evolution of alkaline phosphatase which is less inhibited by its phosphate by-product [42,43]. Similarly, Zhang et al. have shown the use of a femtoliter array combined with a microcapillary for selective recovery of genes expressing improved alkaline phosphatase [44].

##### Optical Methods > Micro Scale Methods > Direct FACS Sorting of Cells

A simpler option, if available, is to dispense with emulsions and directly sort intact cells. This requires ensuring that the fluorescent signal generated by the enzyme remain associated with the cell carrying the cognate gene. Broadly speaking, there are three types of mechanisms in the literature to enable this to happen: (1) Having a surface exposed enzyme whose fluorescent product is retained on the cell surface via covalent or non-covalent methods, (2) an intracellular enzyme with fluorogenic product which is unable to diffuse out of the cell, and (3) a more sophisticated mechanism using fluorescent proteins driven by transcription factors responsive to specific ligands to detect the activity of enzymes which produce (or degrade) said ligands.

Paradigm 1 is exemplified by Olsen et al., who used the adsorption of a positively charged FRET (Forster Resonance Energy Transfer) substrate on the negatively charged cell surface to isolate, using FACS, an OmpT outer membrane protease mutant with altered specificity [45]. The same group later reported other examples [46,47,48] of this approach. Pitzler et al. [49] reported an interesting variation wherein the presence of a hydrolase-glucose oxidase coupled activity leads to radical formation and polymerization of a fluorescent monomer on the surface of *E. coli*, leading to a “fur-shell” structure which can be FACS sorted. This method was then utilized for phytase evolution.

The Liu lab demonstrated a more generally adaptable example of this class by displaying a Sortase A library on the cell surface of yeast *S. cerevisiae*, followed by the exogenous attachment of a Sortase A acceptor peptide. Thereafter, a biotinylated donor peptide was added to the yeast cell suspension to enable fast Sortase A variants to conjugate it to the above-mentioned acceptor peptide. Staining of the donor peptide with Streptavidin-phycoerythrin allowed for more active variants to be isolated by FACS (see schematic in Figure 1E) [50,51].

The second approach is used for intracellular enzymes whose substrates can be expressed in vivo or provided externally and the products of which remain within the cell for FACS sorting, as exemplified by Aharoni et al. [52] and Yang et al. [53]. However, this method is dependent on the presence of a cell permeable substrate and impermeable product, which is rarely available.

The third concept starts by adapting an existing analyte sensitive transcription factor or fluorescent reporter protein, or by engineering an existing biosensor [54] to be specific for the analyte of interest. This in vivo transcription factor then provides a fluorescent signal proportional to the amount of the analyte present by transcribing a fluorescent protein and can be used to monitor the activity of a biosynthetic enzyme which produces the analyte of interest. This concept has been recently demonstrated by Della Corte et al. [55], among others. Meister et al. [56] use a variation which involves the in vivo expression of a fluorescent protein fused to an aggregation inducing peptide separated by a protease site. Cleavage of the non-standard protease site by a suitable mutant coxsackievirus 3C protease allows the reporter to remain soluble, thus enabling detection by FACS. In an interesting variation of this theme, Michener and Smolke [57] have demonstrated the use of analyte responsive RNA switches linked to GFP mRNA translation in yeast to enable the detection of the product theophylline, which was then exploited to evolve improved variants of caffeine demethylase. Other groups have also made use of this strategy to evolve or discover enzymes capable of catalyzing desired reactions [58,59].

An important technical challenge with micro scale methods is DNA retrieval from single sorted cells. As the mutant library is generally present on a multi-copy plasmid, there are multiple copies of each mutant gene per cell. Some researchers report direct transformation of recovered DNA into high efficiency electrocompetent cells [25,30,34]. In some cases, the host cells are still viable after sorting and can be directly placed in suitable medium for growth [50]. PCR to recover the mutant gene sequence is another common approach [44] but may require multiple nested PCRs and/or gel purification to retrieve the desired product.

In sum, micro-scale optical methods have shown excellent potential for enzyme evolution with very high throughput. Some representative schematics are shown in Figure 1.

**Figure 1 molecules-26-05599-f001:**
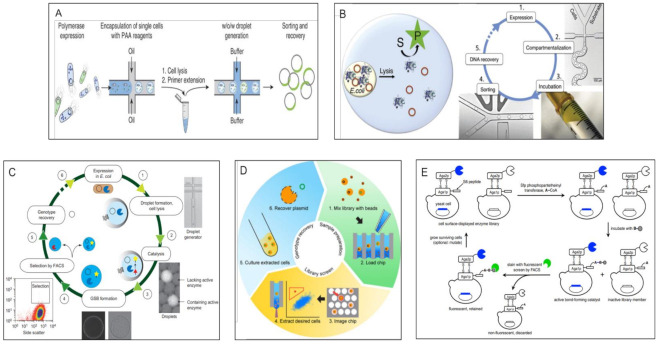
Micro-scale optical methods for enzyme evolution. (**A**) Schematic adapted from Larsen et al. [25] showing the generation of water-in-oil-in-water droplets with cell encapsulation, FACS sorting and plasmid recovery. (**B**) Schematic from Kintses et al. [30] showing the use of a water-in-oil emulsion sorting device for isolation of fluorescent cells expressing active mutants. (**C**) Capture of mutant expressing plasmids and fluorescent substrate by an inducible solid polymer matrix to retain genotype and phenotype linkage prior to FACS for gene recovery. Reproduced from Fischlechner et al. [40]. (**D**) Steps involved in the use of the µscale microcapillary array for identification and isolation of cells expressing active mutants. Reproduced from Chen et al. [42]. (**E**) Use of cell surface display of enzyme and substrate, followed by bond-formation, fluorescent staining and FACS sorting. Reproduced from Chen et al. [50]. Permission was obtained from relevant publisher before reproducing the above figures.

#### 3.1.2. Optical Methods > Macro-Scale Methods

In some cases where technical requirements do not allow for convenient adoption of the micro-scale methods listed above, or where the number of mutants to be screened is lower, it might be more expeditious to use macro-scale (i.e., naked-eye visible reaction scales) methods such as screening in 96 or 384 wells or directly on culture plates. These methods require relatively simpler instrumentation and allow a more flexible assay at the cost of lower throughput. Some common formats are described below.

##### Optical Methods > Macro Scale Methods > Well Plate-Based Methods

96/384 plate-based screening methods are popular and accessible due to their simplicity and quantitative read-out. Broadly, the mutant library is transformed into *E. coli* and plated on a culture plate. Individual colonies are then inoculated into a 96 well plate (higher density plates generally give poor *E. coli* growth due to aeration issues) and grown in a plate shaker, followed by induction, lysis if needed and then assayed with an optical read-out such as fluorescence or absorbance to indicate activity (see Figure 2A) [60,61,62,63]. One issue that must be addressed with plate-based formats is whether the cells need to be lysed, as in the case of intracellular enzymes and non-permeable substrates, which is usually the case. In cases where lysis is needed, options such as a water-bath sonicator or the use of permeabilizing reagents such as detergents (BugBuster or SoluLyse etc.) or anti-microbial peptides (e.g., Cecropin A, polymyxin B etc.), enzymes such as lysozyme, physical processes like freeze–thaw cycles, and autolysis strains such as XJb or combinations thereof are available [64]. The presence of endogenous enzymes from the cell lysate can interfere with the expressed enzyme assay. We found that simple dilution could sometimes be used to ameliorate this interference [61]. If this does not work, purification of the expressed protein using affinity tags may be required. Tags such as GFP or split GFP may also be used to normalize the amount of expressed protein to obtain a specific activity value, which helps resolve confounding factors such as differential expression levels [65,66]. High density plates with 1536 or more wells, and robotic instrumentation for liquid handling or colony picking are highly useful for well-plate based screening; however, with higher density than 96 wells, poor culture aeration becomes a limiting factor (personal observation).

##### Optical Methods > Macro Scale Methods > Culture Plate Based Screening

A simpler option which may be used when a qualitative read-out suffices, and substrate accessibility is not an issue, is culture-plate based screening. This involves growing cell colonies on an agar plate with suitable media, followed optionally by plate replication onto agar plates, or nitrocellulose or Durapore membranes in cases where the protein may be toxic or a white background may be desired, and finally protein expression and exposure to substrate under controlled conditions. Secreted enzyme mutants or a permeable substrate then allow for the generation of an optically distinguishable signal “halo”, highlighting colonies expressing outstanding mutants with larger halos [67,68,69,70]. In other cases, the signal remains in vivo, and the whole colony becomes colored or fluorescent. For instance, O’Loughlin et al. devised a screen for HIV protease mutants with altered specificity by looking for proteases capable of cleaving a β-galactosidase protein reporter, which results in decreased blue coloration in the presence of X-gal, a chromogenic substrate of β-galactosidase [71]. A total of 60,000 variants were reported to have been screened in a single round, highlighting the high throughput and ease of process of plate-based methods. The use of dyes to highlight colonies secreting cellulases has been well attested [72,73].

Further refinements can include the use of freeze-thaw, to release intracellular enzymes, and a “filter-sandwich” format, which allows for more complex multi-step assays with the released enzyme [74,75]. Kermekchiev et al. [76] successfully selected a cold-sensitive Taq polymerase from a total of 3800 mutants using a culture-plate derived filter assay. Weiß et al. [77] used a sophisticated process to screen transaminase variants using a plate based coupled assay, resulting in a colored colony in the presence of the desired specificity (see Figure 2B).

In summary, culture-plate based optical screening allows for relatively high throughput (~10^5^–10^6^) and simple and convenient assays, albeit at the cost of precision. However, where semi-quantitative or qualitative read-outs and moderate throughput suffice, optical plate-based assays are a competitive option owing to their low cost and low equipment needs.

**Figure 2 molecules-26-05599-f002:**
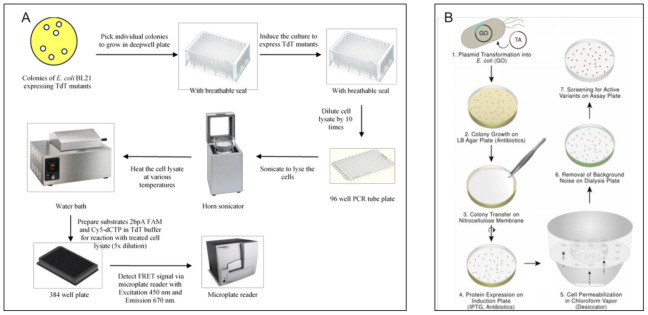
Macroscopic optical methods for enzyme evolution: (**A**) Schematic of the processes involved in a typical 96 well plate based optical assay. Adapted with publisher’s permission from Chua et al. [61]. (**B**) Plate based screening using chromogenic substrate to identify promising mutants. Reproduced with permission from [77].

### 3.2. Survival//Retrieval Based Methods

Another common concept used for enzyme evolution is to link the desired performance of the enzyme to survival or retrieval of the cognate cell or gene. This method, which has multiple formats as described below, can be particularly useful under certain circumstances.

#### 3.2.1. Survival/Retrieval Based Methods > Cell Survival Based Methods

The most direct case is when the cell is unable to survive due to a lack of essential proteins or metabolites or due to the presence of toxic agents in the milieu. The assay is arranged in such a way that only the desired mutant of interest can relieve this blockage to growth and thus selectively allow for survival of only the cell carrying the desired enzyme variant (see schematic in Figure 3A). The upside of this methods is its simplicity and potentially very high throughput; variants are simply subjected to conditions that the original host cannot survive, and survivors are retrieved for verification. The drawback is that it is not possible to link enzyme activity to cell survival in many cases, and the simple survival/death read-out does not allow for fine activity distinctions. Nonetheless, under the right conditions, survival-based screening is unmatched in terms of speed and convenience.

For instance, Hecky and Muller [78] evolved a thermostable variant of beta-lactamase by intentionally destabilizing beta-lactamase with a small C terminal truncation, which leads to decreased ampicillin resistance, expressing this in *E. coli* and selecting for mutants that restore resistance, and then re-attaching the truncated amino acids. This dramatically increased beta-lactamase thermostability as the restoration of the C terminus combined synergistically with the stabilizing effect of the new mutations. It is easy to foresee the same idea being applied to other essential enzymes to improve their thermostability and/or activity.

Park et al. [79] similarly used a survival based selection process to engineer lactamase activity into an αβ/βα metallohydrolase scaffold of glyoxalase II enzyme by modification of active site proximal loops and selection of mutants which could survive in the presence of cefotaxime. Numerous groups have used strains with deletions of metabolic enzymes as hosts to re-evolve the lost activity into a new protein scaffold. For instance, Jurgens et al. [80] selected a Thermotoga maritima N′-[(5′-Phosphoribosyl)formimino]-5-aminoimidazole-4-carboxamide ribonucleotide isomerase (HisA) variant which could also carry out phosphoribosylanthranilate isomerization by transforming a HisA library into an *E. coli* strain with a TrpF deletion, plating the transformants on minimal media lacking tryptophan and isolating plasmids from the survivors. Similar results have been demonstrated with other enzymes [81,82,83,84] with up to 10^6^ mutants screened simply by plating on suitable media.

A particularly interesting application of survival-based selection is the adaptation of modified tRNA and their respective synthetases as orthogonal pairs translating amber stop codons in a different host organism. Such a modified pair can then be used to incorporate modified amino acids with functionality that is absent in the natural amino acids, such as alkynes for click chemistry, phosphate groups for studying the effect of site-specific phosphorylation, and so on [85,86,87,88,89]. In one illustrative example, [90] to evolve such an orthogonal tRNA and synthetase pair, both positive and negative selections are carried out in vivo using a toxic protein (e.g., barnase) and an antibiotic resistance protein (e.g., β-lactamase), both of which carry an amber STOP codon. A tRNA_CUA_ library is expressed without its cognate tRNA synthetase in an *E. coli* host which also carries a barnase gene with an Amber STOP codon at a non-essential position. If a given tRNA_CUA_ is recognized by an *E. coli* tRNA synthetase, it will suppress the Amber STOP codon in barnase, leading to cell death. Thus promiscuous tRNA_CUA_s which can interact with *E. coli* synthetase will be lost. After this step, a positive selection is carried out where the remaining tRNA_CUA_ variants are expressed in the presence of the foreign (*M. jannaschii*) tRNA synthetase as well as a lactamase gene with an Amber STOP codon in a non-essential position. A functional tRNA_CUA_ and tRNA synthetase pair will suppress the Amber codon and permit full length lactamase protein translation, allowing for cell survival in the presence of ampicillin, thus creating an orthogonal pair. A recent interesting application of the survival strategy was demonstrated by Gaudelli et al. [91] who used a survival based selection to evolve a tRNA adenine deaminase (TadA from *E. coli*) to work on a DNA substrate for genome editing purposes by introducing a disabling G to A mutation in antibiotic resistance genes and looking for TadA mutants capable of reversing the same.

Some directed evolution campaigns have used this method for negative selection, that is, to get rid of non-specific clones which form the vast majority of most mutant libraries by linking the said non-specific activity to transcription of a lethal protein [54] or to proteolytic cleavage of essential *E. coli* proteins [71], followed by a more fine grained screening of the remaining library members for positive activity using more quantitative methods like FACS. Such a hybrid approach leverages the best features of plate-based selection (highly parallel but non-quantitative selection) and FACS (quantitative but serial screening).

#### 3.2.2. Survival/Retrieval Based Methods > In Vitro DNA Retrieval Based Methods

Another class of methods which falls in the survival or retrieval-based selection category is Compartmentalized Self-Replication (CSR). It is emulsion based like the FACS/FADS based microdroplet methods discussed above, but instead of an optical read-out, in CSR typically polymerase mutants expressed in *E. coli* are isolated in emulsion droplets along with dNTPs and suitable primer in a suitable buffer and tasked with amplifying their own genes, akin to PCR, under challenging conditions like high temperatures, the presence of inhibitors like blood or mismatched primers (see Figure 3B) [92,93,94], or heavily modified nucleotides [95]. Ong et al. [96] demonstrated a variant of CSR called short patch CSR (spCSR) wherein PCR oligos amplify only a short mutagenized region of the polymerase gene instead of the whole ORF, thus retaining variation information but avoiding the strenuous requirement of amplifying long stretches of nucleic acid. Using spCSR, the authors evolved a polymerase with simultaneous DNA and RNA polymerase activity as well as reverse transcriptase activity.

A further variant is Compartmentalized Self-Tagging (CST) [97,98,99], in which a biotinylated primer binds weakly to plasmids released after heatshock lysis from cells in emulsion. If the cognate polymerase is able to incorporate modified nucleotides into this primer, the strength of binding improves, allowing for robust pulldown of the relevant plasmid onto Streptavidin beads. CST has been used for the incorporation of modified bases such as Hexose Nucleic Acid (HNA), cyclohexenyl nucleic acids (CeNA), Locked nucleic acids (LNA) and others.

An interesting modification of this concept uses ribosome display to link a protein and its cognate mRNA together, followed by compartmentalization and reverse transcription of the mRNA by the cognate enzyme, followed by recovery.

#### 3.2.3. Survival/Retrieval Based Methods > Phage Pulldown Retrieval Methods

In many cases, it is necessary to undertake multiple rounds to reach a level or type of activity desired. The Liu group have developed a method called Phage Assisted Continuous Evolution (PACE) [100] in which multiple rounds of selection are undertaken without user intervention. Briefly, M13 phages carrying mutant genes are selected on the basis of their ability to induce transcription of the essential gene III in the host cell, enabling assembly of viable phage particles. The produced phage particles also undergo mutagenesis owing to the presence of a “mutagenesis plasmid” in the host *E. coli*, generating new diversity at each round of host cell infection. This method was used to evolve T7 RNA polymerases [100], Bacillus thuringiensis (Bt) toxin [101], tRNA synthetases [102], and Cas9 with an altered protospacer adjacent motif (PAM) recognition sequence [103] among others.

Apart from PACE, phage display is more commonly used for discovery of binding moieties like peptides or proteins. However, phage display has also been used for enzyme evolution in other contexts. Many different paradigms have been demonstrated, such as selection of antibodies binding to transition state analogs, which therefore may be able to act as catalysts or “Abzymes” [104], and selection of thermostable enzymes based on resistance to protease cleavage [105] or polymerases based on the ability to catalyze the addition of a biotinylated modified nucleotide onto a locally bound substrate (Figure 3C) [106]. Interestingly, similar results have also been demonstrated by ribosome display as well, which holds the potential for much greater sequence space exploration [107,108,109].

Some illustrative survival/retrieval based methods are depicted in Figure 3.

**Figure 3 molecules-26-05599-f003:**
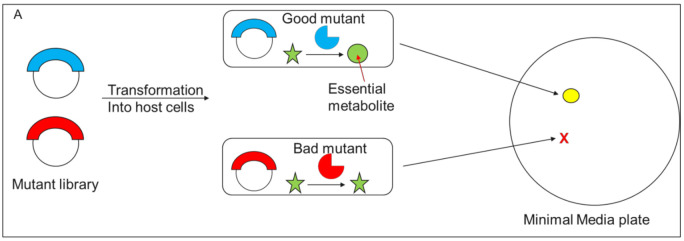
Survival/retrieval based selection: (**A**) A simple schematic showing the use of a mutant library to complement a defective host on minimal media. Only the good mutant can catalyze the generation of an essential metabolite, leading to the formation of a colony. (**B**) Compartmentalized self-replication schematic depiction showing the steps involved. Reproduced with permission from Ghadessy et al. [92]. (**C**) Phage display mediated selection of active polymerase. Polymerases which can incorporate a biotinylated modified nucleotide into a local substrate are able to be pulled down on a solid phase. Reproduced from Chen et al. [106] with permission.

## 4. Conclusions

From the first attempts over 40 years ago [110] to modern methods capable of very high throughput and quantitative readouts, enzyme evolution/screening techniques have come a long way and are now able to screen over a billion mutants. Nonetheless, given the immense diversity present in even small enzymes, there is a great deal of space left to explore and, therefore, enzymes of great value left to discover. In particular, there is room for further miniaturization to single enzyme level, which should allow a significant improvement in throughput and cost, much as moving from 96 well plate assays to emulsions did. Although ribosome or mRNA display technologies might be expected to provide this capability, at this time, these methods have yet to become widespread to the same extent as CSR or optical emulsion sorting. It falls to today’s researchers to come up with new solutions that can continue to improve our ability to locate these diamonds in the rough.

## Data Availability

Not applicable.

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
