# Peer review of "Directed Evolution Methods for Enzyme Engineering"

_molecules, 2021, doi:10.3390/molecules26185599_

Round 1

Reviewer 1 Report

In this manuscript, various methods for enzyme engineering were summarized and reviewed. In general, the manuscript showed the comprehensive summary on the title topic. It needs the following revision before publication in Molecules.

1.The title should be narrowed, since some methods concerning the computation-based rational design for enzyme evolution were not introduced.

2.Latin text should be italicized, like E.coli.

3.The logic for the classification of the enzyme engineering methods should be improved for better illustration.

Author Response

The reviewer's feedback is much appreciated. I've cleaned up the text and inserted clarifications as needed. These changes are highlighted in the Word document review mode.

Answers to specific points are given below:

1.The title should be narrowed, since some methods concerning the computation-based rational design for enzyme evolution were not introduced.

Response: I tried to modify the title to clarify this, but it made the title quite awkward and unwieldy. Instead, I've inserted the words "wet-lab" in the abstract to clarify that the review deals only with wet lab methods.

2.Latin text should be italicized, like E.coli.

Response: This has been done, sorry for the oversight.

3.The logic for the classification of the enzyme engineering methods should be improved for better illustration.

Response: I've tried to clarify the logic of the classification in the introduction passage. In addition, the term "survival/growth" to describe a class of methods has been changed to "survival/retrieval" to better conform to the mechanisms of the methods.

Reviewer 2 Report

In this review  author demonstrates several directed evolution methods for enzyme engineering. This is important topic because adaptation of existing enzymes allows their usage for industrial, research and medical uses. Many methods have been developed over decades and new engineering enzymes  show improved yield, specificity, stability etc.

The manuscript is divided on to several sections and this division is well thought out.  The part 1 (Generation of Diversity to Explore Sequence Space) is well described with proper explanation. However part 2.1.1 (Micro-scale methods) is not so good. In this part methods are described unclear because the author wants to explain a lot in only one sentence. Please add broader explanation with connection to the proper figure. This note is general to the entire manuscript - there is no proper explanation of the figures - only one reference at the end of each section. This make it difficult to understand the methodology. And while macro-scale methods descriptions (part 2.1.2 ) as well as  Survival/Growth Based Methods (part 2.2) are better to understand, the references to the relevant figures would make it  easier for the reader to find sense in the particular method.

Additionally it is difficult for the reader to find how to connect the enzyme of interest at the end of process with its sequence. For macro-methods it is easier to understand but for micro-scale methods where the particular enzyme is  in one drop is unclear. It should be described for all methods – the review is for people to learn.

Minor:

There is error in the first word of abstract , it should be : “Enzymes”

Author Response

I've taken note of the reviewer's comments and made the following changes:

1) The text has been cleaned up and edited for better clarity. The changes are visible in the review mode.

2) Pointers have been inserted to connect the reader to the correct figure when discussing specific methods in the text. 

3) A paragraph has been added at the end of the micro-scale methods section to clarify DNA retrieval methods for single cells, where it is particularly difficult due to the small quantity available.

Round 2

Reviewer 2 Report

The authors made the changes in the current version according to the suggestions.

Author Response

Thanks to the reviewer for the helpful feedback.